# Measuring individual semantic networks: A simulation study

**Samuel Aeschbach**[1,2]*, **Rui Mata**[2], **Dirk U. Wulff**[1,2]

1 Center for Adaptive Rationality, Max Planck Institute for Human Development, Berlin, Germany,
2 Center for Cognitive and Decision Sciences, University of Basel, Basel, Switzerland

* aeschbach@mpib-berlin.mpg.de

**Data availability statement:** All code and information to reproduce the simulation data and analyses is available from GitHub at https://github.com/samuelae/individual-network-simulation. Please find

## Abstract

Accurately capturing individual differences in semantic networks is fundamental to advancing our mechanistic understanding of semantic memory. Past empirical attempts to construct individual-level semantic networks from behavioral paradigms may be limited by data constraints. To assess these limitations and propose improved designs for the measurement of individual semantic networks, we conducted a recovery simulation investigating the psychometric properties underlying estimates of individual semantic networks obtained from two different behavioral paradigms: free associations and relatedness judgment tasks. Our results show that successful inference of semantic networks is achievable, but they also highlight critical challenges. Estimates of absolute network characteristics are severely biased, such that comparisons between behavioral paradigms and different design configurations are often not meaningful. However, comparisons within a given paradigm and design configuration can be accurate and generalizable when based on designs with moderate numbers of cues, moderate numbers of responses, and cue sets including diverse words. Ultimately, our results provide insights that help evaluate past findings on the structure of semantic networks and design new studies capable of more reliably revealing individual differences in semantic networks.

## Introduction

Semantic representations are the basis of central cognitive processes, such as language comprehension and production, reasoning, and problem-solving [1,2]. Understanding the contents and structure of semantic representations and how these change as a function of experience, maturation, and aging are therefore important to advancing our knowledge about a wide range of cognitive phenomena. Two main approaches have emerged that aim to estimate the contents and structure of semantic representations. The first infers semantic representations from massive amounts of text, whereas the second uses behavioral data, such as free associations or semantic judgments [1,3,4]. Semantic representations from text are typically generated by exposing neural networks or other kinds of vector-space models to large amounts of text [4,5]. One benefit of obtaining semantic representations from text is

details on additional, publicly available, third-party source data needed for reproduction at https://github.com/samuelae/ individual-network-simulation/blob/ e5b2e26427c4445b81228a69e5e502f21cf3a57e/ 00_Cold_Storage/01_Source/readme.md.

**Funding:** This work was supported by a grant (197315) from the Swiss National Science Foundation (https://www.snf.ch/en) to Dirk U. Wulff. The funder did not play any role in the study design, data collection and analysis, decision to publish, or preparation of the manuscript.

**Competing interests:** The authors have declared that no competing interests exist.

the scale of available data and the size of the resulting representation. Being based on vast amounts of text, text-based semantic representations typically cover the full body of a language's vocabulary, which permits far-reaching generalizations [5]. However, text is badly suited to capture individual or group differences in semantic representations as information about the authors of used text in model training is often not available. Moreover, text may not represent the ideal source of semantic information for extracting rich mental representations because of the pragmatic communication rules of written expression [5,6]. Behavioral elicitation methods, in turn, are easily implemented at the individual level, and the acquired data can be used to construct semantic representations for specific individuals or groups [6]. Past work has successfully demonstrated this approach using a variety of behavioral paradigms such as free association [7–9], semantic relatedness judgments [10,11], and verbal fluency [10,12]. Recent work suggests, however, that this approach may be subject to significant limitations [9–12], which we discuss below. In this article, we seek to inform future work aiming to measure individual semantic representations by conducting a large-scale recovery simulation to identify the conditions under which key characteristics of semantic networks can be measured with low bias, high resolution, and good generalizability. Our work contributes to clarifying the methodology used to recover individual semantic representations from behavioral data and informing the design of future empirical studies aiming to capture individual differences in semantic representations.

## Estimating semantic networks from behavioral data

Semantic networks are representations of knowledge in the form of a graph, where words are represented by nodes and their relations by edges [1]. Semantic networks can be constructed using data from different behavioral paradigms, including free association and relatedness judgments. The paradigm of free associations presents participants with cue words and asks them to produce one or more spontaneous associations. All other things being equal, the associations are assumed to be produced based on the proximity to the cue in participants' mental representations [13], and thus provide information about the representation's organization. This assumption is in line with the idea that retrieval from semantic representations, or memory in general, follows a spatially constrained search and random walk process, whereby proximate entities are much more likely retrieved than distal ones [14–16]. As a consequence, when prompted to freely retrieve associations to the cue 'cat,' the dominant associations are 'dog,' 'feline,' 'meow,' and 'purr' (based on SWOW-EN [17]), all concepts closely related to 'cat.' The paradigm of relatedness judgments presents participants with two words simultaneously and asks them to judge their relatedness on a numerical scale [10]. Similar to free associations, relatedness judgments are thought to provide signals of the participant's mental representation via the assumption of a semantic decision model, such as the spread of activation or evidence accumulation [18–20].

Evidence from past work investigating group differences in semantic networks suggests that these methods are able to capture relevant individual and group differences in semantic representations. For instance, a number of studies investigating age differences using various behavioral paradigms, including free associations and relatedness judgments, revealed systematic differences as a function of age [8–10,21,22]. Specifically, older adults' networks exhibited structural properties that differed from those of younger adults, such as lower connectedness and clustering [8–10]. Past research has further found semantic network differences between people with different levels of creativity [23], knowledge [24], and intelligence [25].

However, recent research also suggests that methods used in those studies suffer from important psychometric limitations. First, empirical estimates of semantic networks' structural properties likely suffer from biases due to data aggregation and retrieval mechanisms [10]. Past work analyzing data from individual-level relatedness judgments found a network from aggregated responses to overestimate the average degree, clustering coefficient, and average shortest path lengths relative to individual-level networks (see analyses in supplementary material of [10]). Other work analyzing fluency data [12] has shown that inference mechanisms employing unsuitable retrieval mechanisms introduced distortions in the estimation of key network measures, highlighting how retrieval mechanisms can introduce biases, even when measuring networks at the individual level [21,26]. These biases from aggregation and retrieval mechanisms present serious challenges to comparing results between studies.

Second, estimates of individual-level semantic networks likely suffer from low resolution, that is, they lack precision and are only weakly correlated with the true network parameters [9]. Attempting to predict individual differences in semantic tasks in the lab, past work has found individual-level networks to perform no better than a network from aggregate data [9]. This result was attributed to collecting a single rather than multiple sets of responses per cue, limiting the precision of individual-level network estimates and, in turn, resolution. Another study comparing semantic networks inferred from multiple types of behavioral data from the same individuals found that network-wide measures are often unreliable, negatively affecting resolution [11]. However, resolution is critical to deriving solid conclusions about group and individual differences, which has been a core interest in the analysis of semantic networks [1,21].

Third, many results concerning the structure of semantic representations are based on highly specific contents in the mental representation, such as animals, tools, or foods (e.g., [11,12]), potentially limiting their generalizability. The semantic representations of any specific content are potentially structured differently from other contents or the semantic representations as a whole [9]. The highly concrete words in such specific categories (e.g., animals, countries) likely further compound this problem. Concreteness interacts with many aspects of cognitive performance [27], including word retrieval from memory [28]. Insights from semantic networks of concrete words might therefore not generalize. However, generalizability determines both the feasibility of cost-effectively constructing semantic networks and the broader applicability of findings from previous empirical semantic network research.

These limitations present major challenges for the assessment of individual semantic representations from behavioral paradigms. We propose to address these challenges by using simulation methods to explore the extent to which different behavioral paradigms can be used to obtain unbiased, high-resolution, and generalizable estimates of semantic representations. Simulation methods allow us to address these questions by providing full control over the ground truth of the structure and content of semantic representations as well as the design characteristics of behavioral paradigms (e.g., number of items, sample size). With this study, our goal is to advance the methodology of estimating semantic representations from behavioral data and help guide future empirical work by informing the choice of behavioral paradigms and design characteristics that increase future studies' power to estimate individual differences in semantic networks.

## A simulation approach to measuring individual semantic networks

We conduct a large-scale recovery simulation to assess the psychometric limitations of empirical efforts to estimate semantic networks from behavioral data. Our simulation focuses on two often-used paradigms with distinct cognitive underpinnings—free associations

and relatedness judgments—and implements cognitively plausible mechanisms to generate responses from individualized ground-truth semantic networks. We systematically vary important study design dimensions, including the number and similarity of cue words and the number of responses, covering study designs used in past empirical work. For each design configuration, we generate behavioral responses and infer semantic networks, and then compare inferred networks' structural measures against those of the individualized ground truths. This comparison evaluates the suitability of study designs with respect to three key criteria, each one addressing one of the three limitations discussed above. First, are estimates of structural measures unbiased, such that average inferred network measures match the structural measures of the ground truth rather than under- or overestimating them? Second, do the estimates of structural measures have good resolution, such that differences in inferred measures correlate with ground-truth network differences? Third, are estimates of structural measures generalizable, such that network measures of an observed sub-network match those of the whole ground-truth network?

## Materials and methods

Fig 1 presents an overview of our recovery simulation setup. It consists of four main elements: (1) The creation of individualized ground-truth networks based on a pre-trained vector-space

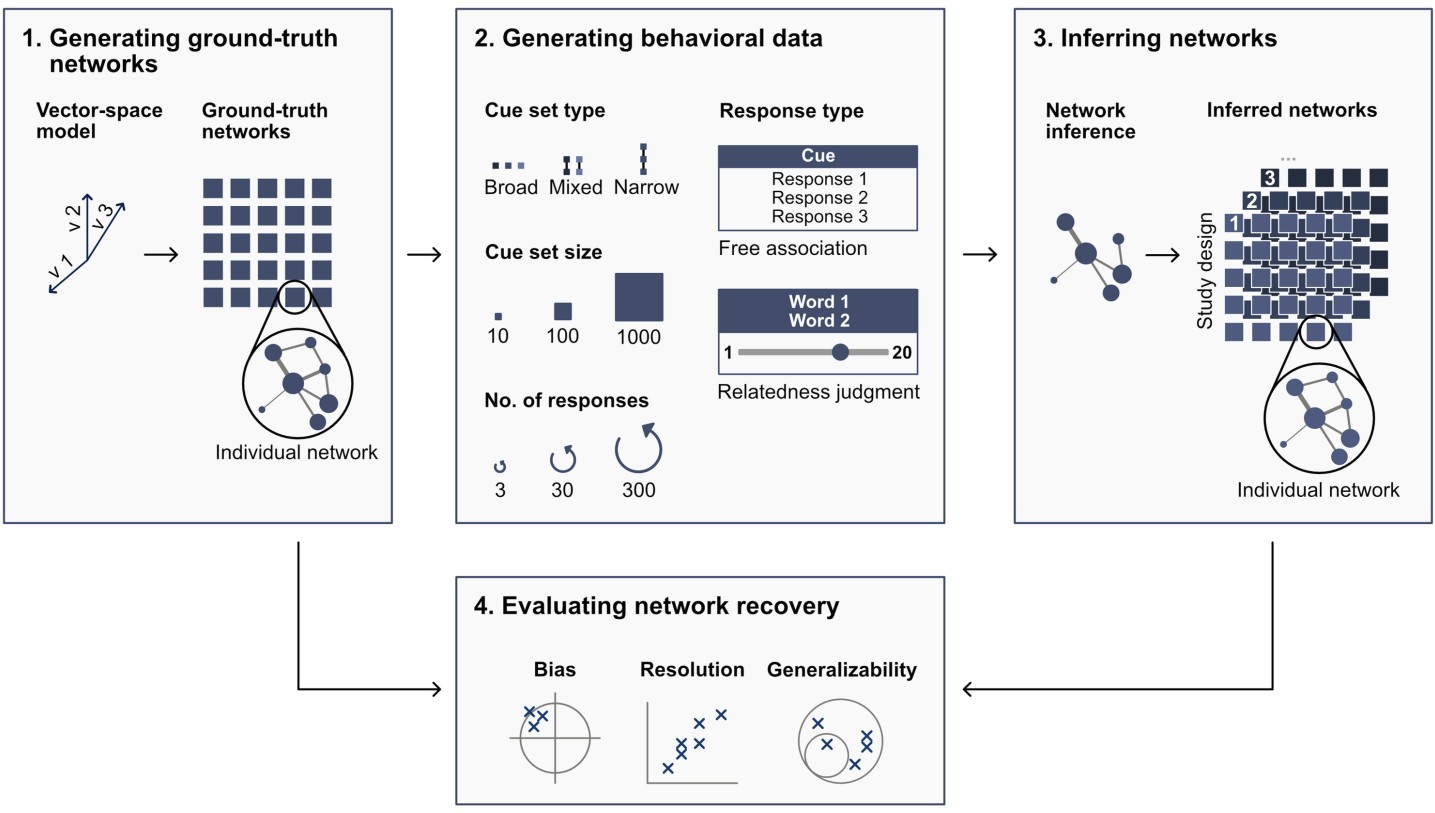

**Fig 1. Overview of the recovery simulation setup**. The simulation setup consists of four elements. (1) Generating individually different ground-truth semantic networks from a vector-space model (i.e., fastText [29]). (2) Simulating behavioral data, varying the study design parameters cue set type, cue set size, number of responses, and response type. (3) Inferring networks from the simulated behavioral data with network inference methods matching the response type. (4) Evaluating the network recovery by comparing the inferred networks' measures to those of the ground-truth networks with respect to bias, resolution, and generalizability.

model, (2) the simulation of behavioral data in the form of free associations and relatedness judgments under different study designs, (3) the inference of semantic networks from the simulated behavioral data, and (4) the comparison of structural network measures between the inferred and ground-truth semantic networks. In the following, we introduce these elements in more detail.

## Generating individualized ground-truth networks

We generated individualized ground-truth networks in two steps. In the first step, we generated a common ground-truth network based on a pre-trained fastText word embedding model [30]. The fastText embedding has been trained on 600 billion tokens of the Common Crawl (https://commoncrawl.org/) and provides, for 2 million words, word embedding vectors that have been shown to accurately predict human behavior [29]. From the fastText embedding, we extracted vectors of 13,486 words that are cues or frequent responses ($n \geq 30$) in the English Small World of Words (SWOW) free association study data [17]. We selected these words for two reasons: to reduce computational complexity and to be able to validate the generation of behavioral data using the publicly available SWOW data. Using this subset, we constructed a common ground-truth network. This is a fully connected network with 13,486 words as nodes and the cosine similarities between the words' embedding vectors as weights of undirected edges.

In the second step, we generated 250 individualized semantic networks from the common ground-truth network. To achieve this, we developed a graph perturbation algorithm that generates networks varying structurally in strength and clustering. See Sect *Network measures* for definitions. The algorithm is inspired by the growth model proposed by Steyvers and Tenenbaum (2005) [31]. Our algorithm uses a triangle score $ts_e = \sum_t^T \prod_i^3 \omega_i$ for each edge $e$, which is determined by the product of each edge weight $\omega_i$ of each edge in each triangle $t$ summed over all triangles $T$ the edge $e$ is part of. Using this triangle score allows us to vary the average clustering of the network independently of its average strength by specifically removing and relocating high and low triangle score edges' weights using the following algorithm:

1. Sample half of the network's edges as sources. With proportion $p$ draw sources from edges with above-median triangle score and with proportion $1-p$ from below-median triangle score edges.
2. Sample edge weight fraction $k \sim \text{Uniform}(0,1)$ for all source edges.
3. For part $1-r$ of all source edges, reduce the edge weight by the previously determined fraction $k$.
4. For the remaining part $r$ of all source edges, relocate the previously determined fraction $k$ of the edge weight to a target edge from the network's other half of non-source edges. Sample target edges based on their current edge weight.
5. Remove all edges with edge weight $\omega < .2$.

We generated individualized networks by running the algorithm 10 times for each of 25 combinations of $p \in [0, 0.75]$ and $r \in [0, 1]$. The 25 combinations have been selected such that the networks' average strength and clustering were uncorrelated ($r = .009$), facilitating an independent evaluation of these two measures. The resulting 250 ground-truth networks serve as a model of individuals with structurally different semantic representations. S1 Table shows a summary of network measures for each parameter value pair.

## Simulating behavioral data

To simulate behavioral data from the individualized semantic networks, we used cognitively plausible retrieval mechanisms of free associations and relatedness judgments tuned to publicly available data. By using these networks as the basis to simulate behavioral data from, the resulting simulated data are a product of the individually different semantic networks, analogous to behavioral data recorded from humans that is based on each person's individually different semantic representation. This work did not involve collecting human data; no ethics approval was required.

**Free association.** Free associations are generated by a retrieval process based on semantic similarity given by the edge weights and word frequency. Past work has shown that local (specific to prior word activation) and global (non-specific to prior word activation) word properties are important to predict the retrieval of words from memory [32]. Specifically, the combination of semantic similarity (local) and word frequency (global) has been found to account well for human memory retrieval [16,32,33]. We implement the process behind a free association response $i$ for cue $j$ as

$$P(i|j) = \frac{w_{ij}^{\gamma_w} \times f_i^{\gamma_f}}{\sum_i w_{ij}^{\gamma_w} \times f_i^{\gamma_f}} \tag{1}$$

where $w_{ij}$ is the edge weight between nodes $i$ and $j$, $f_i$ is the relative word frequency of response $i$ in the SUBTLEXus word frequency database [34], and $\gamma_w$ and $\gamma_f$ are two sensitivity parameters, controlling the process' sensitivity to similarity and frequency, respectively.

We tuned the sensitivity parameters using the SWOW free association database [17]. Specifically, we simulated free association responses for every cue in SWOW using $\gamma_w \in [5, 17.5]$ and $\gamma_f \in [0.25, 1.5]$. We then compared the distribution of responses between simulated and human free associations. Values of $\gamma_w = 10$ and $\gamma_f = 1$ best reproduced the empirical data. See S1 Fig for details.

**Relatedness judgment.** Relatedness judgments are generated based on truncated normal distributions around the corresponding nodes' edge weights. This assumes a minimal process in which relatedness judgments directly reflect semantic similarity within a judge's mental representation, translated into somewhat noisy ratings on a scale from 1 to 20 (following Wulff et al., 2022 [10]). Formally, we implemented the process behind a relatedness judgment for words $i$ and $j$ as

$$RJ(i,j) \sim \mathcal{TN}(w_{ij}^{\gamma}, \sigma^2, 1, 20) \tag{2}$$

where $w_{ij}$ is the edge weight between nodes $i$ and $j$, $\gamma$ is a sensitivity parameter for similarity, and $\sigma$ is the standard deviation of the truncated normal distribution.

We tuned $\gamma$ and $\sigma$ to the MEN relatedness judgment data, a large, publicly available dataset of relatedness judgments [35]. Varying $\gamma \in [1, 10]$ and $\sigma \in [0, 7.65]$, we identified parameters best reproducing the reported inter-rater correlation in the MEN data. Values of $\gamma = 1$ and $\sigma = 3.85$ best reproduced the empirical data. See S2 Fig for details.

**Design configurations.** We generated behavioral data for a number of design configurations to evaluate their impact on network recovery. We varied the cue set size, cue set type, and number of responses across values covering those of past studies.

We varied cue set sizes between 10, 100, and 1,000 cues, implying a maximum network size of 1,000 nodes. This range matches the scale of most studies comparing networks between groups and individuals (e.g., [9–11]) but is does not reach the size of our ground-truth

networks and studies assessing language-level aggregate networks (e.g., SWOW [17]). This range restricts the analysis to economically conceivable designs and limits computational requirements.

We varied the cue set type between *narrow*, *broad*, and *mixed*. Narrow cue sets focus on one semantic topic, reflecting the use of single semantic categories, such as animals or countries (e.g., [10]). Using the common ground-truth semantic network, we generated narrow cue sets by randomly drawing a starting node and successively adding, until the target cue set size is reached, neighboring nodes with the highest average edge weight to previously included nodes. Broad cue sets cover the network widely, reflecting studies aimed at mapping large portions of people's semantic representation (e.g., [9,36]). We generated broad cue sets by randomly drawing a starting node and successively adding random neighbors to the previously added node until the cue set size is reached. Given an average shortest path length of 1.87 (unweighted), this process quickly covers the entire diameter of the network. Finally, mixed cue sets combine elements of narrow and broad cue sets, reflecting the use of cue sets consisting of groups of closely related words (e.g., [23,37]). We generated mixed cue sets by first drawing $n_b = \lfloor \sqrt{N} \rfloor$ nodes according to the broad procedure, where $N$ is the target cue set size. Then, using the broad cues as starting nodes, we generate $N - n_b$ cues following the narrow cue set procedure $n_b$ times. All three cue set types have been used in behavioral studies; however, little is known about how the choice of cue set type influences network characteristics.

Lastly, we varied the number of responses collected per cue between 3, 30, and 300, implying a maximum number of 300,000 responses per estimated network. This range matches the scope of the largest assessments of group-level networks [8] and is orders of magnitude larger than the largest assessments of individual-level networks [7,9].

Overall, this results in design configurations varying on three levels of cue set size, three levels of cue set type, three numbers of responses per cue collected, and two behavioral data types, amounting to 54 design configurations (see also, Fig 1 Panel 2).

## Inferring semantic networks

We inferred semantic networks from the simulated behavioral data of each simulated individual in the following way. For free associations, we set the nodes to the cues of the study design and estimated edges based on the distributions of responses to the cues [5,8]. In detail, we first constructed the cue-response frequency matrix and used it to calculate the positive point-wise mutual information (PPMI) response distributions for each cue. Second, we calculated the cosine similarities between the transformed response distributions. Third, we used the cosine values as the edge weights between cues. This approach represents one of several network inference methods for free associations [7,9,31]. We tested approaches with and without the PPMI transformation and with and without dimensionality reduction of the response distribution. These approaches did not affect the results qualitatively. We did not consider other approaches, such as including responses as nodes. Doing so would have led to varying numbers of nodes and different node sets across networks, severely complicating comparisons between networks.

For relatedness judgment, we also set the nodes to the cues of the study design and estimated edge weights directly from the relatedness judgments. In line with past work [10,11,35], we re-scaled the judgments to the range of 0 to 1. If the number of responses exceeds the number of cue pairs in the cue set, multiple judgments for the same pair are averaged.

## Network measures

To evaluate network recovery, we selected six network measures that are frequently used in the literature and have been linked to cognitive capacities [1,21]. Two of these measures permit within-network comparisons: *edge weight* and *node strength*. The four remaining measures permit between-network comparisons: *average strength*, *average shortest path length (ASPL)*, *average clustering coefficient (average CC)*, and *modularity*. In the following, we briefly introduce each measure. See Barabási and Pósfai (2016) [38] for a detailed introduction.

First, *edge weight* is the collection of edge weights $w_{ij}$ between all nodes $i$ and $j$ in the network. Second, *node strength* is the collection of node strengths defined as $s_i = \sum_j w_{ij}$ where $j$ are all neighbors of node $i$. Third, *average strength* is the average node strength, i.e., Average Strength $= \frac{1}{N} \sum_i s_i$ where $N$ is the number of nodes in the network and $s_i$ the strength of node $i$. Fourth, *ASPL* is the average distance between any two nodes in the network, i.e., ASPL $= \frac{1}{N \cdot (N-1)} \sum_{i \neq j} d_{ij}$, with the distance $d_{ij} = \sum_l 1 - w_{ij,l}$ defined as the sum of 1 minus weight over all $l$ steps of the shortest path. Fifth, *average CC* is the local clustering coefficient averaged over all nodes, i.e., Average CC $= \frac{1}{N} \sum_{i=1}^{N} cc_i$ with the weighted local clustering coefficient of node $i$ defined as $cc_i = \frac{1}{s_i(k_i-1)} \sum_{j,h} \frac{(w_{ij}+w_{ih})}{2} a_{ij} a_{ih} a_{jh}$ where $k_i$ is the number of neighbors of node $i$ and $a$ indicate non-zero weights between the nodes $i$, $j$, and $h$. Sixth, *modularity* is the proportion of edges within node communities determined using the Louvain clustering algorithm [39] relative to the expected proportion under random assignment, i.e., Modularity $= \frac{1}{2m} \sum_{i,j} \left( a_{ij} - \frac{s_i s_j}{2m} \right) \delta(c_i, c_j)$ where $m$ is the number of edges in the network and $\delta(c_i, c_j)$ indicates whether two nodes belong to the same community.

All network measures were calculated using the igraph package version 1.4.3 for R [40]. Except for CC, which we calculate for the 50% strongest edges to avoid ceiling effects, we calculate all network measures using the full network.

## Evaluating network recovery

Our evaluation of network recovery relies on three criteria: bias, resolution, and generalizability. We evaluate each criterion by comparing, for each individual $i$ and each network measure $M$, the measure of their inferred network ($\hat{M}_i$), with that of their ground-truth network ($M_i$). We evaluate bias as the geometric mean of the ratio between the inferred and ground-truth network measures minus one, i.e., Bias $= \left( \prod_{i=1}^{n} \frac{\hat{M}_i}{M_i} \right)^{\frac{1}{n}} - 1$. Biases smaller than zero thus imply smaller values in the inferred than in the ground-truth measures, indicating underestimation. Biases larger than zero, in turn, imply overestimation. We evaluate resolution using Spearman correlations between inferred and ground-truth measures. Both bias and resolution are evaluated in reference to sub-networks of the ground-truth network constrained to the respective study design's cue set. We evaluate generalizability by evaluating bias and resolution in reference to the entire ground-truth network rather than the sub-networks.

All three criteria are evaluated for the four between-network measures, whereas only resolution is evaluated for the within-network measures. There are two reasons for this restriction. First, the assessment of bias for average strength captures all relevant information about the two within-network measures, making an analysis of bias for within-network measures redundant. Second, within-network measures are evaluated at the level of nodes and edges, which is only possible for the same set of nodes in the inferred and ground-truth networks, preventing assessments of generalizability.

We evaluated network recovery across a total of 54 different design configurations consisting of factorial combinations of two response types (free association vs. relatedness judgments), three cue set sizes, three cue set types, and three numbers of responses. Each

design configuration is instantiated 10 times for different sets of cues and completed by 250 synthetic participants. Consequently, each network measure's evaluation of the 54 design configurations is based on 2,500 data points. Overall, the recovery simulation involved the estimation of 135,000 inferred networks based on 30 to 300,000 responses each. The simulations and analyses were performed at the sciCORE scientific computing center at the University of Basel and required approximately 1,500 CPU hours.

## Results

In the following, we present the results of our recovery simulation in terms of three criteria—bias, resolution, and generalizability—that are crucial to evaluating the findings of past studies and designing future, better-informed, studies.

### Evaluating the bias of semantic networks from behavior

First, we report the results on bias, evaluating whether and in which direction inferred and true measures deviate. The ability to infer network measures accurately is important for understanding the general structure of semantic networks and for comparing inferred networks across different methodologies and design configurations. Fig 2 depicts the bias for the four between-network measures. Red tiles and negative values indicate underestimation (inference < ground truth), blue tiles and positive values indicate overestimation (inference > ground truth). Light gray tiles indicate small differences and, thus, low bias. Note that, as bias is determined based on the ratio of inferred and ground-truth network measures, bias values can be interpreted in terms of proportional differences to the ground-truth network measures. To simplify the reporting of the results, we define an acceptable bias level as one within ±30% to the ground-truth measure or, in other words, a bias value between -0.3 and 0.3.

The following noteworthy results emerged. First, our results showed noticeable biases in both directions. Overall, 28.24% of study designs had an acceptable level of bias; the remaining 71.76% study designs under- (29.63%) or overestimated (42.13%) the ground-truth measure, respectively. Second, ASPL was the least biased measure (46.30% of designs with acceptable recovery), followed by modularity (24.07%), average CC (22.22%) and average strength (20.37%). Third, the bias level was generally better for relatedness judgments than for free associations (36.11% vs. 20.37% designs with acceptable recovery), mixed (36.11%) versus broad (29.17%) and narrow (19.44%) cue sets, small (10; 36.11%) versus medium (100; 27.78%) and large (1,000; 20.83%) cue set sizes, and high (300; 48.61%) versus medium (30; 23.61%) and low (3; 12.50%) number of responses.

Fourth, there were noticeable interactions among the design factors and between the design factors and measures. Most notably, average strength had a strong negative bias in free association-based designs irrespective of other design factors, whereas the situation is much better for relatedness judgments. The bad recovery of average strength in free associations can be attributed to a pronounced underestimation of edge weights. This is caused by the fact that, even for large numbers of responses, there is a relatively small chance that two cues produce common responses, which then translates into low cosine similarities and, in turn, low edge weights and strengths. Furthermore, although recovery was better for larger numbers of responses, in relatedness judgments, more responses sometimes increased bias. This can be attributed to the effects of averaging relatedness judgment responses in cases where the design implies multiple responses for every edge in the network, which leads to an overestimation of edges with low or zero weight (cf, regression to mean in probability judgments; [41]). No such distortions affect inferences from free associations.

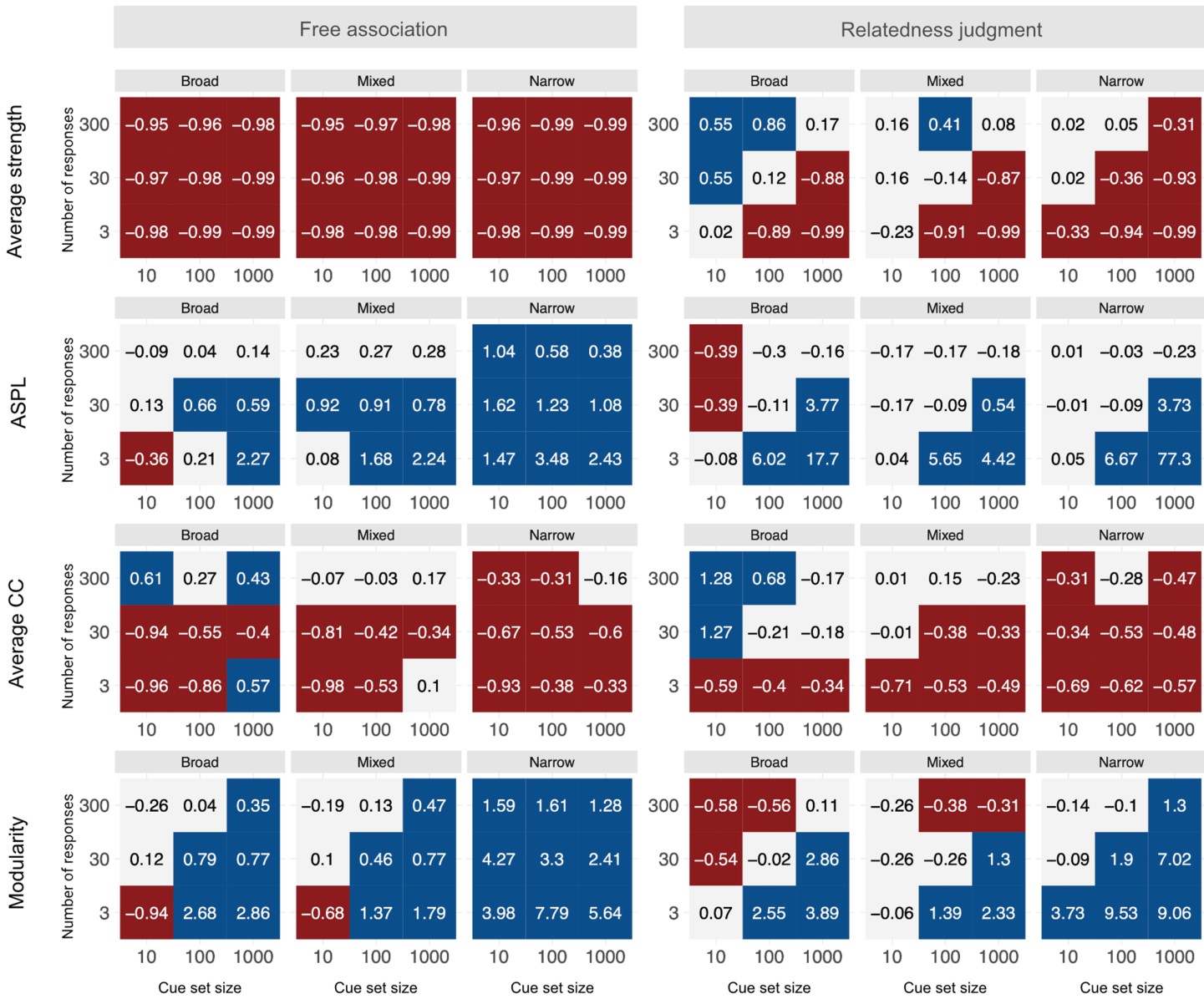

**Fig 2. Bias.** Bias is defined as the ratio between inferred and ground-truth measures. Negative values represent underestimations, whereas positive values represent overestimations. We define an acceptable recovery as bias within ±30% of the ground truth (light gray tiles; −0.3 ≤ bias ≤ 0.3), compared to underestimations of more than 30% (red tiles; bias < −0.3) and overestimation of more than 30% (blue tiles; bias > 0.3).

In sum, several design configurations are capable of recovering ground-truth network characteristics with low bias. Designs that support assessments with low bias are based on mixed and broad cue sets, assess a larger number of responses, and consider detrimental averaging effects in relatedness judgments and the underestimation of average strength in free associations.

## Evaluating the resolution of semantic networks from behavior

Next, we report the results of our analysis of recovery resolution focusing on how well study designs recover differences between the measures of different ground-truth networks. The

higher the resolution of the recovery, the better a study design is suited for comparing semantic networks between individuals and groups. We measure resolution by computing the Spearman correlation ($r$) between the measures of individualized ground-truth and inferred networks. Fig 3 shows the resolution for the two within-network and the four between-network measures. Yellow tiles indicate high resolution, whereas teal and purple tiles indicate poor and negative resolution. To simplify the reporting, we define acceptable recovery as a resolution of $r \geq .5$. A resolution of $r \geq .5$ corresponds to medium-sized true effects (Cohen's $d = 0.5$) in studies with 200 participants per group to be detected in 80% of cases using a one-sided significance test ($\alpha = .05$). See S3 Fig for details.

Several noteworthy findings emerged. First, the resolution of network estimates' recovery is acceptable for a large number of designs (50.00%), poor for 44.44%, negative for 5.25%, and missing in 0.31% due to ceiling effects of clustering. Note that negative resolutions imply inferences opposite to those in the ground truth. As with bias, the resolution varies strongly between design configurations and measures.

Second, resolution was highest for average strength (87.04% of designs have acceptable resolution), followed by ASPL (59.26%), average CC (55.56%), modularity (42.59%), node strength (37.04%), and edge weight (18.52%).

Third, resolution was generally higher for relatedness judgments (62.35%) than for free associations (37.65%), for large numbers of responses (300; 67.59%) versus medium (30; 57.41%) and small (3; 25.00%) numbers of responses, for medium (100; 54.63%) and large (1,000; 51.85%) versus small (10; 43.52%) cue set sizes, and for mixed (54.63%) and narrow (52.78%) versus broad cues set types (42.59%).

Fourth, there were again noticeable interactions among the design factors and between the design factors and measures. The within-network measures (node strength and edge weight) had substantially higher resolution for relatedness judgments as compared to free associations. As with the high bias of average strength, the low resolution of the within-network measures can, in part, be attributed to the underestimation of edge weights in free associations. Furthermore, whereas resolution is generally better with larger cue sets and more responses, irrespective of response type or cue set type, the resolution of edge weight, node strength, and modularity in relatedness judgments is best for small cue sets and many responses. We do not have a good explanation for modularity, but the different patterns of results for within-network and most between-network measures are likely related to different levels of comparison. Cue set size determines the size of the network, which strongly influences the estimation noise at the level of between-network measures but less so at the level of within-network measures. Finally, modularity in free associations and narrow cue set types has bad resolution without apparent systematic influences of other design factors.

In sum, the resolution of semantic network measures is high in many design configurations. For between-network measures, except for modularity, many responses and large cue sets show high resolution in both response types, whereas for within-network measures, higher numbers of responses and smaller cue set sizes lead to the best resolution.

## Evaluating the generalizability of semantic networks from behavior

Finally, we report the results of our analysis of generalizability, assessing bias and resolution in relation to the whole ground-truth network. So far, our analyses have evaluated recovery by comparing inferred measures to those of ground-truth sub-networks including only the words used as cues (local analysis). However, in many cases, it is of interest to make inferences that generalize to the whole network, beyond the set of cues included in the design, to uncover the characteristics of the entire semantic representation. Fig 4 shows bias and

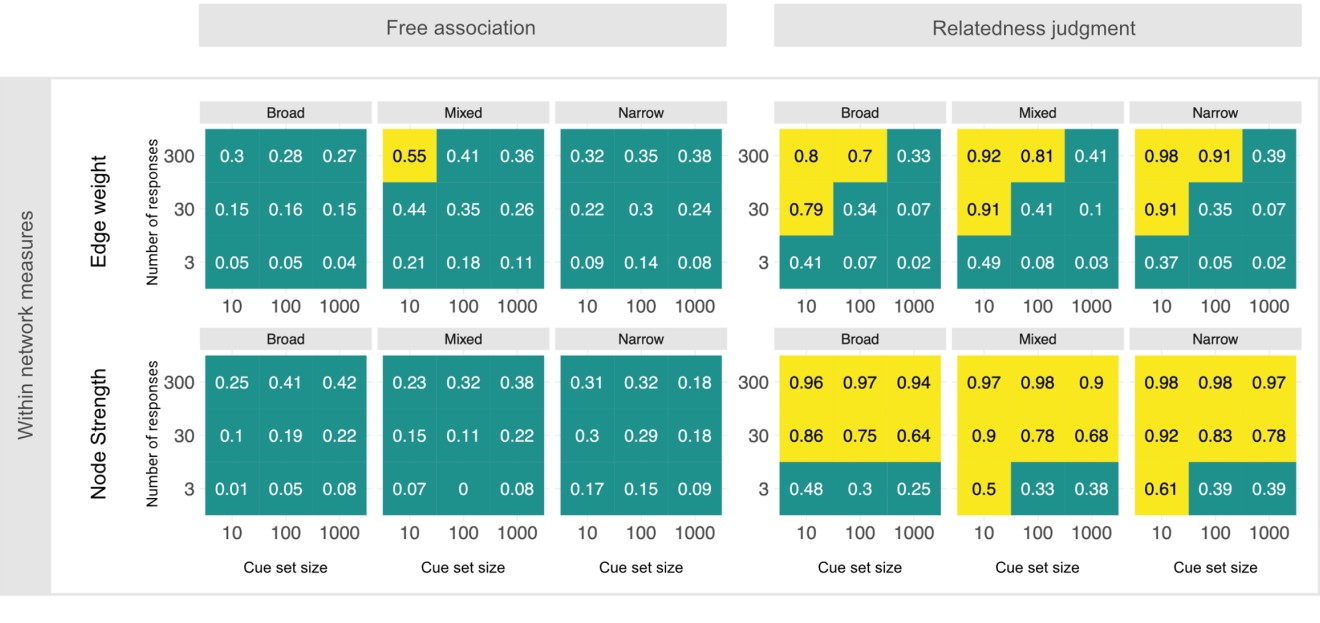

**Fig 3. Resolution.** Reported as Spearman correlations (*r*) between measures of the ground-truth networks and inferred networks. Resolution of the recovery is defined as acceptable for values $r \geq .5$ (yellow tiles), positive for values $0 \leq r < .5$ (teal tiles), and negative for values $r < 0$ (purple tiles). Mean Spearman correlations of inferred metrics and ground-truth metrics.

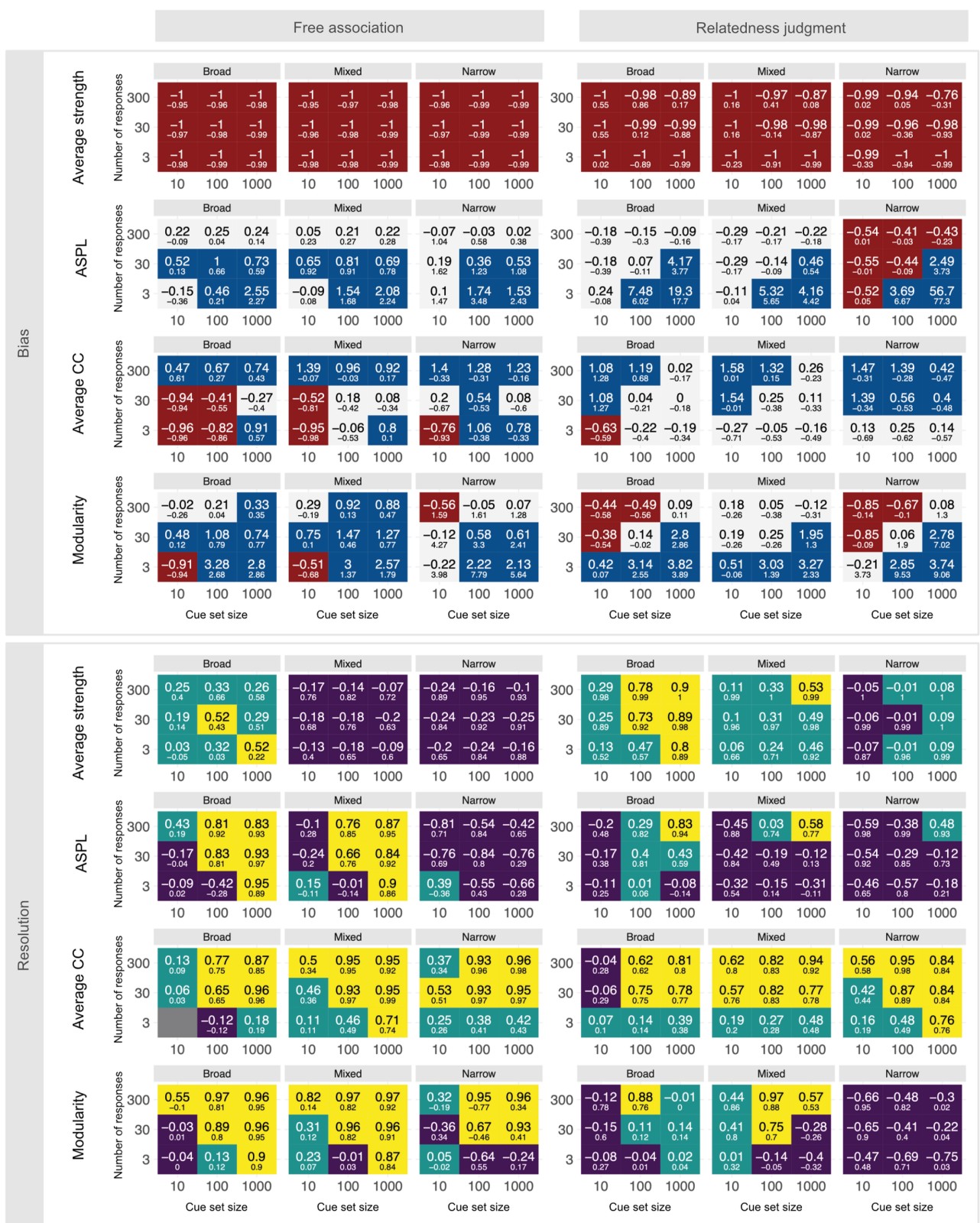

**Fig 4. Generalizability of bias and resolution.** Larger font numbers indicate global comparison values, smaller font numbers indicate local comparison values.

resolution comparing the estimates of measures to those of the entire ground-truth network (global analysis). To simplify the reporting, we define acceptable recovery using the same criteria applied above (–0.3 < bias < 0.3; resolution $\geq$ .5).

The generalizability analysis revealed several noteworthy results. First, generalizability was acceptable in 29.17% of bias assessments and 32.87% of resolution assessments. The result for bias implies comparable recovery to the local analysis reported above (+0.93 percentage points), whereas the result for resolution implies a considerable deterioration (-28.24) as compared to the local analysis.

Second, generalizability varied substantially between measures. Bias was best in assessments of ASPL (44.44% acceptable recovery), followed by average CC (37.037%), modularity (35.19%), and average strength (0%) showing no acceptable recovery at all. In comparison to the local analysis, average strength (-20.37 percentage points) and ASPL (-1.85) worsened, whereas average CC (14.82) and modularity (11.11) improved. Resolution was best in assessments of average CC (57.41% acceptable recovery), followed by modularity (37.04%), ASPL (22.22%), and average strength (14.82%). In comparison to the local analysis, this implied a significant deterioration for average strength (-72.22 percentage points) and ASPL (-37.04), but not modularity (-5.56) and average CC (+1.85).

Third, generalizability also varied between design factors. Recovery bias was generally better for relatedness judgments (33.33% acceptable recovery) than for free associations (25%), but the difference between the behavioral paradigms was reduced compared to local analysis. In contrast to recovery bias, recovery resolution was higher for free association (39.82%) than relatedness judgments (25.93%). Further, design configurations with large number of responses resulted in less bias (300 responses; 38.89% acceptable recovery) as compared to medium (30; 26.39%), and small number of responses (3; 22.22%). Similarly, design configurations with a large number of responses had higher resolution (300; 48.61%) as compared to medium (30; 38.89%) and small numbers of responses (3; 11.11%). These results are consistent with those obtained in the local analysis. Cue set size did not systematically affect global bias, with large (1,000 cues, 29.17%), medium (100; 29.17%), and small cue sets (10; 29.17%) showing equal levels of bias, whereas in the local analysis, smaller cue sets showed less bias. By contrast, cue set size substantially impacted resolution, with large cue set sizes (1,000; 50%) showing better recovery than medium (100; 38.89%) and small cue set sizes (10; 9.72%). These differences were stronger in the global than in the local analysis. Finally, mixed (30.56% good recovery), broad (29.17%), and narrow cue set types (27.78%) showed roughly equal bias, similar to the results of the local analysis. However, broad (38.89%) and mixed (38.89%) cue sets produced higher resolution than narrow cue set types (20.83%). This contrasts with the local analysis, where narrow and mixed cue sets outperformed broad cue set types.

Fourth, there were noteworthy interactions among the design factors. Cue set size and number of responses interacted for relatedness judgments such that a certain ratio led to the lowest bias. This may be explained by the same factor as for local comparisons: with an increasing number of responses, edges are overestimated due to averaging effects, which happens most for small cue sets because the number of possible edges increases non-linearly with increasing cue set sizes. Furthermore, network measures interacted with cue set type, but not consistently across response types. This was especially the case for resolution, where narrow cue sets mostly produced lower resolution than broad cue set types; however, for average CC, all cue set types produced similarly good resolution.

Overall, resolution, but not bias, worsened when comparing the inferred network measures to those of the entire network rather than the sub-network assessed by a given design's cue set. Given that the largest assessed sub-network covered less than one-tenth of the entire network,

the drop in resolution is not surprising. Notwithstanding, generalizability can be achieved for most measures by using broad or mixed cue sets and at least moderate numbers of cues and responses.

## Discussion

Accurately capturing individual differences in semantic networks is fundamental to advancing our mechanistic understanding of changes in semantic memory. Past empirical attempts to measure individual-level networks from behavioral paradigms have shown limitations in terms of bias, resolution, and generalizability. In this study, we carried out a large-scale recovery simulation comparing inferred and ground-truth semantic networks for a variety of network measures and study design configurations. Our analyses revealed that a low-bias and high-resolution assessment of semantic representation is achievable for most measures using either behavioral paradigm. Our analyses also revealed a strong dependency on design factors, including complicated interactions, that can severely limit the psychometric properties of semantic network inference. These findings have implications for interpreting past evidence on semantic network structure, designing future empirical studies, and determining the set of research questions that can and cannot be appropriately assessed using semantic networks from behavioral data. We next discuss four key takeaways in more detail.

The first takeaway is that absolute estimates of network measures should rarely be trusted due to high levels of bias for most design configurations. We found estimates to differ by at least 30% from the true values in 69.44% (local analysis) and 71.30% (global analysis) of cases. Furthermore, biases varied wildly between nearly 100% underestimation and 7,730% overestimation. These biases were not consistently improved by any design factor, except a larger number of responses. These findings have implications for past findings on the structural makeup of semantic networks. For instance, past work has been interested in assessing whether semantic networks exhibit a small-world structure similar to networks in other domains, such as power grids and biological neuronal networks [10,31,42]. Small-world networks are characterized by high clustering and low average shortest path lengths. Because biases for these two network measures vary strongly and can go in opposite directions, conclusions regarding small-world characteristics of semantic networks may be easily distorted. Another implication is that comparisons of semantic networks between behavioral paradigms and design configurations will be impacted by different bias, such that such comparisons between different study designs could potentially be meaningless.

The second key takeaway is that measurements of semantic networks are characterized by high resolution for many study design configurations. As a consequence, meaningful comparisons within a given study design are possible. Resolution tended to be highest when using large cue set sizes and number of responses. However, most of the resolution is achieved with moderate sizes, implying that high-resolution measurements of semantic networks are potentially economically attainable. S4 Fig presents a detailed analysis of the effect of resolution on sample size requirements. The finding that acceptable resolution can be achieved in moderate, more economical study design configurations is crucial for past and future comparisons between the semantic measures of individuals and groups [8–10].

The third takeaway from our simulation is that structural measures of semantic networks based on behavioral responses can generalize to broader semantic representations, given a suitable study design. To support generalizations, study designs should use moderate to large cue sets of mixed or broad cue set type. Results obtained in studies using narrow cue set types, for instance, consisting of members of just one category, such as animals [10], therefore likely do not generalize and are limited to the specific set of cues assessed.

The fourth takeaway is that, despite the recommendations highlighted above, there are no universal patterns strictly favoring one study design configuration. Moreover, the intricate interactions between design factors suggest that our recommendations may not hold in all circumstances. We encourage researchers to carefully consider the priorities in their investigation and to choose design configurations accordingly, potentially using additional, more targeted recovery simulations.

There are several limitations we would like to highlight. First, our general and individualized ground-truth networks are based on assumptions that may be inconsistent with the true representations in people's minds. Our understanding of the structure and content of mental representations is still limited [21]. Moreover, recent work demonstrates that semantic representations derived from different sources—such as text, behavioral, or brain data—systematically diverge in the information they encode, with behavior-based embeddings in particular capturing unique aspects of psychological meaning compared to the text-based ground truth used in our simulation [43]. Together with the challenges highlighted by this investigation, it is currently likely that there are other equally or more justified sets of assumptions. Future research should, therefore, evaluate the recovery of network measures under different ground-truth assumptions, in particular, by using different base representations and algorithms to individualize networks. Second, our simulation assumes constant retrieval processes in the generation of behavioral data. Although this assumption helped us assess the impact of design configurations in a controlled setting, individuals probably differ not only in semantic representations but also in semantic retrieval processes [1,3,9,44]. Future studies should account for this by conducting recovery analyses under varying retrieval process assumptions. Third, some of the bias and lack of resolution in inferred networks can potentially be addressed using different algorithms for network inference. We tested only a small number of options that were suitable for our analysis plan. However, other proposals have been made (e.g., random walk-based measures of similarity [12,45,46]) and should be tested in future work. Fourth and finally, we focused only on some of the most frequently used measures of network structure and omitted several others, such as assortativity or diameter, which could also be studied in future work.

## Conclusion

Our investigation has demonstrated that the structure of semantic representations can be inferred successfully from human behavior when appropriate study designs are used and comparisons are limited to common design configurations. Study designs that support good inference recruit moderate cue set sizes and number of responses, as well as mixed or broad cue set types. All in all, our results suggest that meaningful insights into individual and group differences in the structure of semantic representations can be obtained and guide the design of future empirical work.

## Supporting information

**S1 Table. Individual network generation parameters and network measures**. Parameter pairs for $p$ and $r$ used in generating individual semantic ground-truth networks along with mean values (standard deviations) of network measures for each parameter pair over all 10 iterations. Note that all individualized networks have 13,486 words as nodes.
(PDF)

**S1 Fig. Free association parameter tuning.** (A) Pearson correlations and (B) Spearman rank correlations of the probability distributions of first responses between model free association

responses and SWOW free association norms [17]. Both correlation measures suggest $\gamma_w = 10$ and $\gamma_f = 1$ as best-fitting parameter values with $r_{Pearson} = 0.453$ and $r_{Spearman} = 0.353$. (C) Pearson correlations of first responses analogously to Panels A and B, however, using fixed model parameters $\gamma_w = 10$ and $\gamma_f = 1$, varying semantic network ground truth minimal edge weight. This analysis shows an edge weight cut off of 0.2, as implemented in the study, to perform well in comparison to other cut offs. Free association model (D) first response, (E) second response, and (F) third response word median rank among SWOW norms [17] for the same cue word. Panels D, E, and F show the model parameters $\gamma_w = 10$ and $\gamma_f = 1$, as best-fitting values, to generate monotonously increasing median ranks ($Med(R_1) = 2$, $Med(R_2) = 4$, $Med(R_3) = 5$).
(TIF)

**S2 Fig. Relatedness judgment parameter tuning.** (A) Correlations between model using networks of varying lower cut offs and MEN relatedness judgment norms [35], showing 0.20 to be a good compromise between computational resource sparsity and performance. (B) Correlations between model and MEN norms [35] for varying $\gamma$ values, showing $\gamma = 1$ to be the best fit reproducing the behavioral ratings. (C) Tuning of model $\sigma$ parameter to MEN norms [35] inter-rater reliability of $r_{Spearman} = 0.68$ by simulating raters A and B and correlating their ratings, showing $\sigma = 0.15$ to best approximate the reported inter-rater reliability.
(TIF)

**S3 Fig. Simulation of statistical power for resolution levels.** We simulated the statistical power of studies with a medium-sized true effect of Cohen's $d = 0.5$ at varying levels of resolution and sample size to find a resolution of $r = .5$ to correspond to a power of $1 - \beta = .8$ in studies comparing samples of $n = 200$ using a one-sided t-test at $\alpha = .05$.
(TIF)

**S4 Fig. Simulation of statistical power for sample sizes.** We simulated the statistical power of studies with a medium-sized true effect of Cohen's $d = 0.5$ at varying levels of resolution and sample sizes to find sample sizes promoting to a power of $1 - \beta \geq .8$ for study designs with resolutions $r = .61$ and $r = .40$ using a one-sided t-test at $\alpha = .05$.
(TIF)

## Acknowledgments

We thank Laura Wiles for editing the manuscript. Calculations were performed at sciCORE (http://scicore.unibas.ch/) scientific computing center at University of Basel.

## Author contributions

**Conceptualization:** Samuel Aeschbach, Rui Mata, Dirk U. Wulff.

**Data curation:** Samuel Aeschbach.

**Formal analysis:** Samuel Aeschbach, Dirk U. Wulff.

**Funding acquisition:** Dirk U. Wulff.

**Investigation:** Samuel Aeschbach, Dirk U. Wulff.

**Methodology:** Samuel Aeschbach, Dirk U. Wulff.

**Project administration:** Samuel Aeschbach.

**Software:** Samuel Aeschbach, Dirk U. Wulff.

**Supervision:** Rui Mata, Dirk U. Wulff.

**Visualization:** Samuel Aeschbach, Dirk U. Wulff.

**Writing – original draft:** Samuel Aeschbach, Rui Mata, Dirk U. Wulff.

**Writing – review & editing:** Samuel Aeschbach, Rui Mata, Dirk U. Wulff.

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
