## [Decision Letter · Decision Letter 0]

24 Mar 2025

PONE-D-24-48310Measuring individual semantic networks: A simulation studyPLOS ONE

Dear Dr. Aeschbach,

Thank you for submitting your manuscript to PLOS ONE. After careful consideration, we feel that it has merit but does not fully meet PLOS ONE’s publication criteria as it currently stands. Therefore, we invite you to submit a revised version of the manuscript that addresses the points raised during the review process.

We look forward to receiving your revised manuscript.

Kind regards,

Bruno Alejandro Mesz, Ph.D.

Academic Editor

PLOS ONE

Journal Requirements:

2. Please note that funding information should not appear in the Acknowledgments section or other areas of your manuscript. We will only publish funding information present in the Funding Statement section of the online submission form. Please remove any funding-related text from the manuscript. 

**Additional Editor Comments:**

Two experts in the field have carefully reviewed the manuscript entitled “Measuring individual semantic networks: A simulation study“. You can find their comments below. They both had very positive comments on the manuscript but also requested clarifications and elaborations of some parts. 

In light of these reviews, I am requesting a minor revision and resubmission, in which you will need to respond to each point in each review. 

Reviewers' comments:

Reviewer's Responses to Questions

**Comments to the Author**

1. Is the manuscript technically sound, and do the data support the conclusions?

Reviewer #1: Yes

Reviewer #2: Yes

2. Has the statistical analysis been performed appropriately and rigorously? 

Reviewer #1: Yes

Reviewer #2: Yes

3. Have the authors made all data underlying the findings in their manuscript fully available?

Reviewer #1: Yes

Reviewer #2: Yes

4. Is the manuscript presented in an intelligible fashion and written in standard English?

Reviewer #1: Yes

Reviewer #2: Yes

5. Review Comments to the Author

Reviewer #1: Reviewer background: statistics and linguistics

Summary: The work is a simulation study that aims to address limitations in measuring individual differences in semantic networks. They focus on constructing individual semantic networks from two behavioral paradigms: free associations and relatedness judgement tasks. Their goal is to identify certain design configurations of these semantic networks that makes comparisons between individual networks accurate and meaningful. They start by building individual ground-truth networks from a vector space model, and then they generate behavioral data from which they construct many different configurations of inferred individual semantic networks. They then compare the inferred networks with the ground truth networks to assess bias, resolution, and generalizability of network measurements. These criteria allow the researchers to identify the design configurations in which it is suitable/unsuitable to compare individual semantic networks in a meaningful way.

Overall impression: The paper is well-written and easy to understand. It is a very useful study for researchers who work with individual semantic networks, as it can contribute to more methodologically sound investigations. The results are interesting and can easily be applied. Overall it is a great paper that I think will be of great interest to a specific group of researchers.

Suggestions: Some parts of the introduction could benefit from more details. Also, some of the methodology is somewhat unclear and could benefit from more detailed explanations. I provide more specific suggestions below.

Lines 25-26: “Recent work suggests, however, that this approach may be subject to significant limitations [9–12].” What kinds of limitations?

Lines 61-71: I believe this paragraph could benefit from significant elaboration. This paragraph is essentially the main motivation for the study. I think you should offer more details about the limitations you aim to investigate. Describe the biases, how they emerge, the types of problems they can cause, and give examples. Do the same for resolution and generalizability. By elaborating on these three potential limitations, the reader is more prepared to understand the analyses that you do in the simulation study.

Section “Generating individual ground-truth networks”: You describe how the networks are created, but it could be helpful to the reader to elaborate on the meaning of these ground-truth networks. What do they represent? Does each network represent a hypothetical ground-truth mental-lexicon of an individual person? (This is how I understood it). It could be helpful to provide a summary table of the statistics of the 250 networks (nodes, edges, density, clustering).

Section “Simulating behavioral data”: This part is somewhat unclear to me. Where are the edge weights coming from in equations 1 and 2? Are they coming from the fully connected vector space model ground-truth network? This is what I understood. Also, here it could be helpful to discuss what this data represents. It represents essentially “sampling” from the ground-truth mental lexicons of an individual person, and that sampling is done by asking questions in a behavioral experiment. Therefore the methods you describe to generate these data (free associations and relatedness judgement) are simulations of this sampling (This is how I understand it).

Section “Design configurations”: At the end of this section, I think it would be useful to summarize the number of total configurations you have, which you don’t do until later, in the paragraph immediately before the Results. Here, a summary table could be helpful as well.

Section “Evaluating network recovery”: This part is unclear to me. How are the comparisons between inferred networks and ground-truth networks made? Is one inferred network compared to one ground-truth network? What exactly are M and M-hat? What measures are they? Are they calculated on individual networks, one inferred network of a given configuration and one ground truth sub-network reduced to match that configuration?

Nice paper!!

Reviewer #2: Summary

- This research investigates the methodological challenges in measuring individual semantic networks through behavioural paradigms (of free associations and relatedness judgments). Using comprehensive recovery simulations, the authors evaluate how different study design parameters affect the bias, resolution and generalisability of semantic network measurements. The study certainly makes a significant contribution by systematically assessing the conditions under which semantic networks can be reliably inferred from behavioural data.

Strengths

- The methodology used is rigorous, examining multiple design parameters, with any shortcomings that become apparent reading through being clearly addressed in the limitations and future work. There are also practical implications for future empirical work, making this a useful reference work in the field.

- The evaluation of network measures is comprehensive, although given the part focus on broad, narrow and mixed cue sets, it is regrettable as a reader that remaining matters such as network diameter could not be included in this work.

- The insights on the generalisability of these network measures is a particularly impactful consideration, especially to identify the limitations of previous works.

Issues

- Page 4, Section “Generating individualized ground-truth networks” – There should be a simple justification/validation for the choice of parameter range of `p` and `r` to make obvious there are no complex underlying reasons.

- Page 5, Given that it is the basis of the synthesised responses and further experiments, the cognitive plausibility of the retrieval mechanisms used in the simulations could benefit from stronger theoretical grounding, or perhaps elaboration of the references to justify it. However, this is considered in the discussion, especially mentioning the potentially disparate nature of individual semantic retrieval processes. It could be deemed a necessary simplicity for, as written controllability, when considering that including methodology from Human Modelling/Computational Rationality fields would make this work overly complex.

- Page 9, Figure 3. Resolution caption has an error (assumed typographical) in the range for teal tiles “0 ≥ r < .5” -> “0 <= r < .5”.

Nitpicks (These do not need to be addressed)

- The discussion could address potential limitations of using fastText embeddings as ground truth.

- There was one sentence thought a little vague: “the networks’ average strength and clustering were `largely` independent” perhaps could be more concretely communicated, but it’s easily understood so no need to change.

PLOS ONE 7 Criteria

- Original Research. ✓

- Results not published elsewhere. ✓

- The simulation methodology is rigorous and well-documented. Statistical analyses are appropriate. ✓

- Conclusions supported by data. ✓

- The writing is clear and professional throughout. ✓

- No ethical concerns as this is a simulation study. ✓

- GitHub repository with code and data links for reproduction. ✓

An exceptionally well written paper. The suggested revisions are only to strengthen an already solid piece of work.

Overall recommendation: Accept with Minor Revisions

6. PLOS authors have the option to publish the peer review history of their article (what does this mean?). If published, this will include your full peer review and any attached files.

Reviewer #1: **Yes: **Katherine Elizabeth Abramski

Reviewer #2: **Yes: **Owen G.W. Saunders

---

## [Author Response · Author response to Decision Letter 1]

13 Jun 2025

Dear Dr. Mesz, dear Reviewers,

Thank you for your input and for the opportunity to revise our manuscript, “Measuring individual semantic networks: A simulation study” for publication in PLOS One.

As we detail in the Responses to reviewers attachment, we have addressed all points raised by you and the reviewers. We very much appreciate your and the reviewers’ help with improving our contribution and hope you agree that our improved manuscript is now ready for publication.

In addition, as further requested, I removed all funding information from the Acknowledgments section. Would you be able to include the following statement in the appropriate section instead?

“This work was supported by a grant from the Swiss National Science Foundation to Dirk U. Wulff (197315).”

Kind regards,

Samuel Aeschbach, on behalf of the authors

---

## [Decision Letter · Decision Letter 1]

6 Jul 2025

Measuring individual semantic networks: A simulation study

PONE-D-24-48310R1

Dear Dr. Aeschbach,

We’re pleased to inform you that your manuscript has been judged scientifically suitable for publication and will be formally accepted for publication once it meets all outstanding technical requirements.

Kind regards,

Bruno Alejandro Mesz, Ph.D.

Academic Editor

PLOS ONE

Additional Editor Comments (optional):

Reviewers' comments:

Reviewer's Responses to Questions

**Comments to the Author**

1. If the authors have adequately addressed your comments raised in a previous round of review and you feel that this manuscript is now acceptable for publication, you may indicate that here to bypass the “Comments to the Author” section, enter your conflict of interest statement in the “Confidential to Editor” section, and submit your "Accept" recommendation.

Reviewer #2: All comments have been addressed

2. Is the manuscript technically sound, and do the data support the conclusions?

Reviewer #2: Yes

3. Has the statistical analysis been performed appropriately and rigorously? 

Reviewer #2: Yes

4. Have the authors made all data underlying the findings in their manuscript fully available?

Reviewer #2: Yes

5. Is the manuscript presented in an intelligible fashion and written in standard English?

Reviewer #2: Yes

6. Review Comments to the Author

Reviewer #2: (No Response)

7. PLOS authors have the option to publish the peer review history of their article (what does this mean?). If published, this will include your full peer review and any attached files.

Reviewer #2: No

---

## [Editor Report · Acceptance letter]

PONE-D-24-48310R1

PLOS ONE

Dear Dr. Aeschbach,

I'm pleased to inform you that your manuscript has been deemed suitable for publication in PLOS ONE. Congratulations! Your manuscript is now being handed over to our production team.

Kind regards,

on behalf of

Dr. Bruno Alejandro Mesz

Academic Editor

PLOS ONE